# Test-Time Adaptation to Distribution Shift by Confidence Maximization and Input Transformation

## Abstract

Deep neural networks often exhibit poor performance on data that is unlikely under the train-time data distribution, for instance data affected by corruptions. Previous works demonstrate that test-time adaptation to data shift, for instance using entropy minimization [1], effectively improves performance on such shifted distributions. This paper focuses on the fully test-time adaptation setting, where only unlabeled data from the target distribution is required. This allows adapting arbitrary pretrained networks. Specifically, we propose a novel loss that improves test-time adaptation by addressing both premature convergence and instability of entropy minimization. This is achieved by replacing the entropy by a non-saturating surrogate and adding a diversity regularizer based on batch-wise entropy maximization that prevents convergence to trivial collapsed solutions. Moreover, we propose to prepend an input transformation module to the network that can partially undo test-time distribution shifts. Surprisingly, this preprocessing can be learned solely using the fully test-time adaptation loss in an end-to-end fashion without any target domain labels or source domain data. We show that our approach outperforms previous work in improving the robustness of publicly available pretrained image classifiers to common corruptions on such challenging benchmarks as ImageNet-C.

## 1 Introduction

Deep neural networks achieve impressive performance on test data, which has the same distribution as the training data. Nevertheless, they often exhibit a large performance drop on test (target) data which differs from training (source) data; this effect is known as data shift [2] and can be caused for instance by image corruptions. There exist different methods to improve the robustness of the model during training [3, 4, 5]. However, generalization to different data shifts is limited since it is infeasible to include sufficiently many augmentations during training to cover the excessively wide range of potential data shifts [6]. Alternatively, in order to generalize to the data shift at hand, the model can be adapted during test-time. Unsupervised domain adaptation methods such as [7] use both source and target data to improve the model performance during test-time. In general source data might not be available during inference time, e.g., due to legal constraints (privacy or profit). Therefore we focus on the *fully test-time adaptation* setting [1]: the model is adapted to the target data given only the arbitrarily pretrained model parameters and the unlabeled target data that share the same label space as source data. We extend the work of Wang et al. [1] by introducing a novel loss function, using a diversity regularizer, and prepending a parametrized input transformation module to the network. We show that our approach outperform previous works and make pretrained models robust against common corruptions on image classification benchmarks as ImageNet-C [8] and ImageNet-R [9].

Sun et al. [10] investigate test-time adaptation using a self-supervision task. Wang et al. [1] and Liang et al. [11] use the entropy minimization loss that uses maximization of prediction confidence as self-supervision signal during test-time adaptation. Wang et al. [1] has shown that such loss performs

better adaptation than a proxy task [10]. When using entropy minimization, however, high confidence predictions do not contribute to the loss significantly anymore and thus provide little self-supervision. This is a drawback since high-confidence samples provide the most trustworthy self-supervision. We mitigate this by introducing two novel loss functions that ensure that gradients of samples with high confidence predictions do not vanish and learning based on self-supervision from these samples continues. Our losses do not focus on minimizing entropy but on minimizing the *negative log likelihood ratio* between classes; the two variants differ in using either soft or hard pseudo-labels. In contrast to entropy minimization, the proposed loss functions provide non-saturating gradients, even when there are high confident predictions. We refer to Figure 1 for an illustration of the losses and the resulting gradients. Using these new loss functions, we are able to improve the network performance under data shifts in fully test-time adaptation.

In general, self-supervision by confidence maximization can lead to collapsed trivial solutions, which make the network to predict only a single or a set of classes independent of the input. To overcome this issue a *diversity regularizer* [11, 12] can be used, that acts on a batch of samples. It encourages the network to make different class predictions on different samples. We extend the regularizer by including a moving average, in order to include the history of the previous batches and show that this stabilizes the adaptation of the network to unlabeled test samples. Furthermore we also introduce a parametrized *input transformation module*, which we prepend to the network. The module is trained in a fully test-time adaptation manner using the proposed loss function, i. e. without the need of any target domain labels or source data. It aims to partially undo the data shift at hand. This helps to further improve the performance on image classification benchmark with corruptions.

Since our method does not change the training process, it allows to use any pretrained models. This is beneficial because any good performing pretrained network can be readily reused, e.g., a network trained on some proprietary data not available to the public. We show, that our method significantly improves performance on models that are trained on clean ImageNet data such as a ResNet50 [13], as well as robust models such as ResNet50 models trained using DeepAugment+AugMix [9].

In summary our main contributions are as follows: we propose non-saturating losses based on the negative log likelihood ratio, such that gradients from high confidence predictions still contribute to test-time adaptation. We extend the diversity regularizer that acts on a batch of samples to a moving average version, which includes the history of the previous batch samples. This prevents the network from collapsing to trivial solutions. Furthermore we also introduce an input transformation module, which partially undoes the data shift at hand. We show that the performance of different pretrained models can be significantly improved on challenging benchmarks like ImageNet-C and ImageNet-R.

## 2   Related work

**Common image corruptions** are potentially stochastic image transformations motivated by real-world effects that can be used for evaluating a model's robustness. One such benchmark, ImageNet-C [8], contains simulated corruptions such as noise, blur, weather effects, and digital image transformations. Additionally, Hendrycks et al. [9] proposed three data sets containing real-world distribution shifts, including Imagenet-R. The ImageNet-C have been further extended to MNIST [14], several object detection datasets [15], and image segmentation [16], reflecting the interest of the robustness community. Most proposals for improving robustness involve special training protocols, requiring time and additional resources. This includes data augmentation like Gaussian noise [17, 18, 9], CutMix [19], AugMix [4], training on stylized images [3, 20] or against adversarial noise distributions [21]. Mintun et al. [22] pointed out that many improvements on ImageNet-C are due to data augmentations which are too similar to the test corruptions, that is: overfitting to ImageNet-C occurs. Thus, the model might be less robust to corruptions not included in the test set of ImageNet-C.

**Unsupervised domain adaptation** methods train a joint model of the source and target domain by cross-domain losses, with the hope to find more general and robust features. These losses optimize feature alignment [23, 24] between domains, adversarial invariance [25, 5, 26, 27], shared proxy tasks [28] or adapting the entropy minimization via an adversarial loss [7]. While these approaches are effective, they require explicit access to source and target data at the same time, which may not always be feasible. Our approach works with any pretrained model and only needs target data.

**Test-time adaptation** (also termed *source free adaptation* in some literature) is a setting, when training (source) data is unavailable at test-time. Several works use generative models [29, 30, 31, 32]

for the source free adaptation and require several thousand epochs to adapt to the target data [30, 32]. Besides, there is another line of work [10, 33, 34, 35, 1] that interpret the common corruptions as data shift and aim to improve the model robustness against these corruptions with efficient test-time adaptation strategy to facilitate online adaptation. Such setting refrain the usage of generative models or methods that require larger number of adaptation steps. Our work also falls in this line of research and aims to test-time adapt the model to common corruptions with less computational overhead.

Sun et al. [10] update feature extractor parameters at test-time via a self-supervised proxy task (predicting image rotations). However, Sun et al. [10] alter the training procedure by including the proxy loss into the optimization objective as well, hence arbitrary pretrained models cannot be used directly for test-time adaptation. Inspired by the domain adaptation strategies [36, 37], several works [33, 34, 35] replace the estimates of Batch Normalization (BN) activation statistics with the statistics of the corrupted test images. Fully test time adaptation, studied by Wang et al. [1] (TENT) uses entropy minimization to update the channel-wise affine parameters of BN layers on corrupted data along with the batch statistics estimates. SHOT [11] also uses entropy minimization and a diversity regularizer to avoid collapsed solutions. SHOT modifies the model from the standard setting by adopting weight normalization at the fully connected classifier layer during training to facilitate their pseudo labeling technique. Hence, SHOT is not readily applicable to arbitrary pretrained models.

We show that pure entropy minimization [1, 11] results in vanishing gradients for high confidence predictions, thus inhibiting learning. Our work addresses this issue by proposing a novel non-saturating loss, that provides non-vanishing gradients for high confidence predictions. We show that our proposed loss function improves the network performance after test-time adaptation. In particular, performance on corruptions of higher severity improves significantly. Furthermore, we add and extend the diversity regularizer [11, 12] to avoid collapse to trivial, high confidence solutions. Note that the existing diversity regularizers [11, 12] act on a batch of samples, hence the number of classes has to be smaller than the batch size. We mitigate this problem by extending the regularizer to a running average version. Prior work [5, 38, 39] transformed inputs by an additional module to overcome domain shift, obtain robust models, and also to learn to resize. In our work, we also prepend an input transformation module to the model, but in contrast to former works, this module is trained purely at test-time to partially undo the data shift at hand and thus aids the adaptation.

# 3 Method

We propose a novel method for fully test-time adaption. For this, we assume that a neural network $f_\theta$ with parameters $\theta$ is available that was trained on data from some distribution $\mathcal{D}$, as well a set of (unlabeled) samples $X \sim \mathcal{D}'$ from a target distribution $\mathcal{D}' \neq \mathcal{D}$ (importantly, no samples from $\mathcal{D}$ are required). We frame fully test-time adaption as a two-step process: (i) Generate a novel network $g_\phi$ based on $f_\theta$, where $\phi$ denotes the parameters that are adapted. A simple variant for this is $g = f$ and $\phi \subseteq \theta$ [1]. However, we propose a more expressive and flexible variant in Section 3.1. (ii) Adapt the parameters $\phi$ of $g$ on $X$ using an unsupervised loss function $L$. We propose two novel losses $L_{slr}$ and $L_{hlr}$ in Section 3.2 that have non-vanishing gradients for high-confidence self-supervision.

## 3.1 Input Transformation

We propose to define the adaptable model as $g = f \circ d$. That is: we preprend a trainable network $d$ to $f$. The motivation for the additional component $d$ is to increase expressivity of $g$ such that it can learn to (partially) undo the domain shift $\mathcal{D} \to \mathcal{D}'$.

Specifically, we choose $d(x) = \gamma \cdot [\tau x + (1 - \tau)r_\psi(x)] + \beta$, where $\tau \in \mathbb{R}$, $(\beta, \gamma) \in \mathbb{R}^{n_{in}}$ with $n_{in}$ being the number of input channels, $r_\psi$ being a network with identical input and output shape, and $\cdot$ denoting elementwise multiplication. Specifically, $\beta$ and $\gamma$ implement a channel-wise affine transformation and $\tau$ implements a convex combination of unchanged input and the transformed input $r_\psi(x)$. By choosing $\tau = 1$, $\gamma = \mathbf{1}$, and $\beta = \mathbf{0}$, we ensure $d(x) = x$ and thus $g = f$ at initialization. In principle, $r_\psi$ can be chosen arbitrarily. In this work, we choose $r_\psi$ as a simple stack of $3 \times 3$ convolutions, group normalization, and ReLUs (for details, we refer to the appendix). However, exploring other choices would be an interesting avenue for future work.

Importantly, while the motivation for $d$ is to learn to partially undo a domain shift $\mathcal{D} \to \mathcal{D}'$, we train $d$ end-to-end in the fully test-time adaptation setting on data $X \sim \mathcal{D}'$, without any access to samples

from the source domain $\mathcal{D}$, based on the losses proposed in Section 3.2. The modulation parameters of $g_\phi$ are $\phi = (\beta, \gamma, \tau, \psi, \theta')$, where $\theta' \subseteq \theta$. That is, we adapt only a subset of the parameters $\theta$ of the pretrained network $f$. We largely follow Wang et al. [1] in adapting only the affine parameters of normalization layers in $f$ while keeping parameters of convolutional kernels unchanged. Additionally, batch normalization statistics (if any) are adapted to the target distribution.

Please note that the proposed method is applicable to any pretrained network that contains normalization layers with a channel-wise affine transformation. Even for networks that do not come with such affine transformation layers, one can add affine transformation layers into $f$ that are initialized to identity as part of model augmentation.

## 3.2 Adaptation Objective

We propose a loss function $L = L_{\text{div}} + \delta L_{\text{conf}}$ for fully test-time network adaptation that consists of two components: (i) a term $L_{\text{div}}$ that encourages predictions of the network over the adaptation dataset $X$ that match a target distribution $p_{\mathcal{D}'}(y)$. This can help avoiding test-time adaptation collapsing to too narrow distributions such as always predicting the same or very few classes. If $p_{\mathcal{D}'}(y)$ is (close to) uniform, it acts as a diversity regularizer. (ii) A term $L_{\text{conf}}$ that encourages high confidence prediction on individual datapoints. We note that test-time entropy minimization (TENT) [1] fits into this framework by choosing $L_{\text{div}} = 0$ and $L_{\text{conf}}$ as the entropy.

### 3.2.1 Class Distribution Matching $L_{div}$

Assuming knowledge of the class distribution $p_{\mathcal{D}'}(y)$ on the target domain $\mathcal{D}'$, we propose to add a term to the loss that encourages the empirical distribution of (soft) predictions of $g_\phi$ on $X$ to match this distribution. Specifically, let $\hat{p}_{g_\phi}(y)$ be an estimate of the distribution of (soft) predictions of $g_\phi$. We use the Kullback-Leibler divergence $L_{\text{div}} = D_{KL}(\hat{p}_{g_\phi}(y) \| p_{\mathcal{D}'}(y))$ as loss term. In a special case of $p_{\mathcal{D}'}(y)$ being a uniform distribution over the classes, this corresponds to maximizing the entropy $H(\hat{p}_{g_\phi}(y))$. Similar assumption has been made in SHOT [11] to circumvent the collapsed solutions.

Since the estimate $\hat{p}_{g_\phi}(y)$ depends on $\phi$, which is continuously adapted, it needs to be re-estimated on a per-batch level. Since re-estimating $\hat{p}_{g_\phi}(y)$ from scratch would be computational expensive, we propose to use a running estimate that tracks the changes of $\phi$ as follows: let $p_{t-1}(y)$ be the estimate at iteration $t-1$ and $p_t^{emp} = \frac{1}{n} \sum_{k=1}^n \hat{y}^{(k)}$, where $\hat{y}^{(k)}$ are the predictions (confidences) of $g_\phi$ on a mini-batch of $n$ inputs $x^{(k)} \sim X$. We update the running estimate via $p_t(y) = \kappa \cdot p_{t-1}(y) + (1-\kappa) \cdot p_t^{emp}$. The loss becomes $L_{\text{div}} = D_{KL}(p_t(y) \| p_{\mathcal{D}'}(y))$ accordingly. We use $\kappa = 0.9$ in the experiments.

### 3.2.2 Confidence Maximization $L_{conf}$

We motivate our choice of $L_{\text{conf}}$ step-by-step from the (unavailable) supervised cross-entropy loss: for this, let $\hat{y} = g_\phi(x)$ be the predictions (confidences) of model $g_\phi$ and $H(\hat{y}, y^r) = -\sum_c y_c^r \log \hat{y}_c$ be the cross-entropy between prediction $\hat{y}$ and some reference $y^r$. Moreover, let the last layer of $g$ be a softmax activation layer $\text{softmax}$. That is $\hat{y} = \text{softmax}(o)$, where $o$ are the network's logits. We note that we can rewrite the cross-entropy loss in terms of the logits $o$ and a one-hot reference $y^r$ as follows: $H(\text{softmax}(o), y^r) = -o_{c^r} + \log \sum_{i=1}^{n_{cl}} e^{o_i}$ where $c^r$ is the index of the 1 in $y^r$ and $n_{cl}$ is the number of classes.

In the case of labels being available for the target domain (which we do not assume) in the form of a one-hot encoded reference $y_t$ for data $x_t$, one could use the *supervised cross-entropy loss* by setting $y^r = y_t$ and using $L_{sup}(\hat{y}, y^r) = H(\hat{y}, y^r) = H(\hat{y}, y_t)$. Since fully test-time adaptation assumes no label information being available, the supervised cross-entropy loss is not applicable and other options for $y^r$ need to be used.

One option are (hard) *pseudo-labels*. That is, one defines the reference $y^r$ based on the network predictions $\hat{y}$ via $y^r = \text{onehot}(\hat{y})$, where $\text{onehot}$ creates a one-hot reference with the 1 corresponding to the class with maximal confidence in $\hat{y}$. This results in $L_{pl}(\hat{y}) = H(\hat{y}, \text{onehot}(\hat{y})) = -\log \hat{y}_{c^*}$, with $c^* = \arg\max \hat{y}$. One disadvantage with this loss is that the (hard) pseudo-labels ignore uncertainty in the network predictions during self-supervision. This results in large gradient magnitudes with respect to the logits $|\frac{\partial L_{pl}}{\partial o_{c^*}}|$ being generated in situations where the network is highly unconfident (see

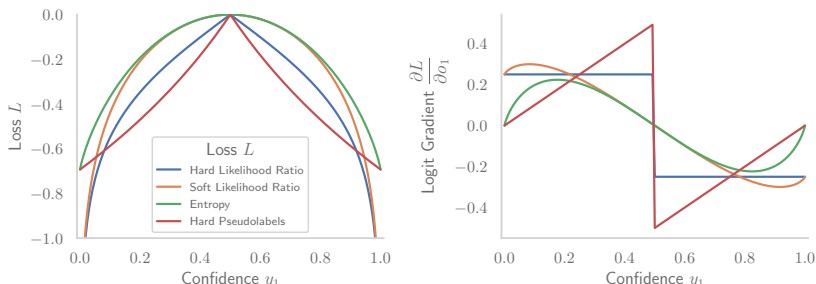

Figure 1: *Illustration of different losses for confidence maximization.* Losses (left, shifted such that maxima of all losses are at 0) and the resulting gradients with respect to the first logit (right) as a function of the first classes confidence are shown for the case of a binary classification problem. Both *entropy* and *hard pseudo-labels* have vanishing gradients for high confidence predictions. Accordingly, both have maximum gradient amplitude for low-confidence self-supervision, with this effect being stronger for the hard pseudo-labels. *Hard Likelihood Ratio* has constant gradient amplitude for any confidence and thus takes into account low- and high-confidence self-supervision equally. *Soft Likelihood Ratio* also shows non-vanishing (albeit non-maximum) gradients for high-confidence self-supervision and additionally produces small gradient amplitudes from low-confidence self-supervision. Since the likelihood ratio-based losses are unbounded, the design of the model needs to ensure that logits cannot grow unbounded.

Figure 1). This is undesirable since it corresponds to the network being affected most by data points where the network's self-supervision is least reliable.

An alternative is to use soft pseudo-labels, that is $y^r = \hat{y}$. This takes uncertainty in network predictions into account during self-labelling and results in the *entropy minimization* loss of TENT [1]: $L_{ent}(\hat{y}) = H(\hat{y}, \hat{y}) = H(\hat{y}) = -\sum_c \hat{y}_c \log \hat{y}_c$. However, also for the entropy the logits' gradient magnitude $|\frac{\partial L_{ent}}{\partial o}|$ goes to 0 when one of the entries in $\hat{y}$ goes to 1 (see Figure 1). For a binary classification task, for instance, the maximal logits' gradient amplitude is obtained for $\hat{y} \approx (0.82, 0.18)$. This implies that during later stages of test-time adaptation where many predictions typically already have very high confidence, i. e. above $0.82$, gradients are also dominated by datapoints with relative low confidence in self-supervision.

While both hard and soft pseudo-labels are clearly motivated, they are not optimal in conjunction with a gradient-based optimizer since the self-supervision from low confidence predictions dominates (at least during later stages of training). To address this issue, we propose two losses that are analogous to $L_{pl}$ and $L_{ent}$, but are not based on the cross-entropy $H$ but instead on the negative log likelihood ratios

$$R(\hat{y}, y^r) = -\sum_c y_c^r \log \frac{\hat{y}_c}{\sum_{i \neq c} \hat{y}_i} = -\sum_c y_c^r (\log \hat{y}_c - \log \sum_{i \neq c} \hat{y}_i) = H(\hat{y}, y^r) + \sum_c y_c^r \log \sum_{i \neq c} \hat{y}_i$$

Note that while the entropy $H$ is lower bounded by 0, $R$ can get arbitrary small if $y_c^r \to 1$ and the sum $\sum_{i \neq c} \hat{y}_i \to 0$ and thus $\log \sum_{i \neq c} \hat{y}_i \to -\infty$. This property will induce non-vanishing gradients for high confidence predictions.

The first loss we consider is the *hard likelihood ratio* loss that is defined similarly to the hard pseudo-labels loss $L_{pl}$:

$$L_{hlr}(\hat{y}) = R(\hat{y}, \text{onehot}(\hat{y})) = -\log(\frac{\hat{y}_{c^*}}{\sum_{i \neq c^*} \hat{y}_i}) = -\log(\frac{e^{o_{c^*}}}{\sum_{i \neq c^*} e^{o_i}}) = -o_{c^*} + \log \sum_{i \neq c^*} e^{o_i},$$

where $c^* = \arg \max \hat{y}$. We note that $\frac{\partial L_{hlr}}{\partial o_{c^*}} = -1$, thus also high-confidence self-supervision contributes equally to the maximum logits' gradients. This loss was also independently proposed as negative log likelihood ratio loss by Yao et al. [40] as a replacement to the fully-supervised cross entropy loss for classification task. However, to the best of our knowledge, we are the first to motivate and identify the advantages of this loss for self-supervised learning and test-time adaptation due to its non-saturating gradient property.

In addition to $L_{hlr}$, we also account for uncertainty in network predictions during self-labelling in a similar way as for the entropy loss $L_{ent}$, and propose the *soft likelihood ratio* loss:

$$L_{slr}(\hat{y}) = R(\hat{y}, \hat{y}) = -\sum_c \hat{y}_c \cdot \log(\frac{\hat{y}_c}{\sum_{i \neq c} \hat{y}_i}) \qquad = -\sum_c \hat{y}_c \log(\frac{e^{o_c}}{\sum_{i \neq c} e^{o_i}})$$

$$= \sum_c \hat{y}_c (-o_c + \log \sum_{i \neq c} e^{o_i})$$

We note that as $\hat{y}_{c^*} \to 1$, $L_{slr}(\hat{y}) \to L_{hlr}(\hat{y})$. Thus the asymptotic behavior of the two likelihood ratio losses for high confidence predictions is the same. However, the soft likelihood ratio loss creates lower amplitude gradients for low confidence self-supervision. We provide illustrations of the discussed losses and the resulting logits' gradients in Figure 1.

We note that both likelihood ratio losses would typically encourage the network to simply scale its logits larger and larger, since this would reduce the loss even if the ratios between the logits remain constant. However, when finetuning an existing network and restricting the layers that are adapted such that the logits remain approximately scale-normalized, these losses can provide a useful and non-vanishing gradient signal for network adaptation. We achieve this appproximate scale normalization by freezing the top layers of the respective networks. In this case, normalization layers such as batch normalization prohibit "logit explosion". However, predicted confidences can presumably become overconfident; calibrating confidences in a self-supervised test-time adaptation setting is an open and important direction for future work.

## 4   Experimental settings

**Datasets** We evaluate our method on image classification datasets for corruption robustness and domain adaptation. We evaluate on the challenging benchmark ImageNet-C [8], which includes a wide variety of 15 different synthetic corruptions with 5 severity levels that attribute to data shift. This benchmark also includes 4 additional corruptions as validation data. For domain adaptation, we choose ImageNet trained models to adapt to ImageNet-R proposed by Hendrycks et al. [9]. This dataset contains various naturally occurring artistic renditions of object classes from the original ImageNet. ImageNet-R comprises 30,000 image renditions for 200 ImageNet classes. Please refer Sec. A.5 for the experiments on other domain adaptation datasets VisDA-C [41], Office-Home [42].

**Models** Our method operates in a fully test-time adaptation setting that allows us to use any arbitrary pretrained model. We use publicly available ImageNet pretrained models ResNet50, DenseNet121, ResNeXt50, MobileNetV2 from torchvision [43]. We also test on a robust ResNet50 model trained using DeepAugment+AugMix [1] [9].

**Baseline for fully test-time adaptation** Since TENT from Wang et al. [1] outperformed competing methods and fits the fully test-time adaptation setting, we consider it as a baseline and compare our results to this approach. Similar to TENT, we also adapt model features by estimating the normalization statistics and optimize only the channel-wise affine parameters on the target distribution.

**Settings** We conduct test-time adaptation on a target distribution for 5 epochs with batch size 64 and use the Adam optimizer with cosine decay scheduler of the learning rate with initial value 0.0006. We set the weight of $L_{conf}$ in our loss function to $\delta = 0.025$ and $\kappa = 0.9$ in the running estimate $p_t(y)$ of $L_{div}$ (we investigate the effect of $\kappa$ in the Sec. A.3). Similar to SHOT [11], we also choose the target distribution $p_{\mathcal{D}'}(y)$ in $L_{div}$ as a uniform distribution over the available classes. We found that the models converge during 3 to 5 epochs and do not improve further.

For TENT, we use SGD with momentum 0.9 at constant learning rate 0.00025 with batch size 64. These values correspond to the ones of Wang et al. [1]; alternative settings of optimizer and learning rates for TENT did not improve performance. TENT is originally optimized only for 1 epoch. For a fair comparison to our method, we optimize TENT also for 5 epochs. Similar to Wang et al. [1], we also control for ordering by data shuffling and sharing the order across the methods.

Note that all the hyperparameter settings are tuned solely on the validation corruptions of ImageNet-C that are disjoint from the test corruptions. As discussed in Section 3.2.2, we freeze all trainable parameters in the top layers of the networks to prohibit "logit explosion". Note that normalization

---

[1] From https://github.com/hendrycks/imagenet-r. Owner permitted to use it for research/commercial purposes.

Table 1: Test-time adaptation of ResNet50 on ImageNet-C at highest severity level 5. Ground truth labels are used to adapt the model in supervised manner to obtain empirical upper bound performance.

| Method | Gauss | Shot | Impulse | Defocus | Glass | Motion | Zoom | Snow | Frost | Fog | Bright | Contrast | Elastic | Pixel | JPEG |
|---|---|---|---|---|---|---|---|---|---|---|---|---|---|---|---|
| No Adaptation | 2.44 | 2.99 | 1.96 | 17.92 | 9.82 | 14.78 | 22.50 | 16.89 | 23.31 | 24.43 | 58.93 | 5.43 | 16.95 | 20.61 | 31.65 |
| Pseudo Labels | 2.44 | 2.99 | 1.96 | 17.92 | 9.82 | 14.78 | 22.50 | 16.89 | 23.31 | 24.43 | 58.93 | 5.43 | 16.95 | 20.61 | 31.65 |
| Epoch 1 | | | | | | | | | | | | | | | |
| TENT | 32.70 | 35.34 | 35.11 | 32.79 | 31.80 | 47.22 | 53.02 | 51.82 | 43.42 | 60.44 | 68.82 | 27.53 | 58.47 | 61.63 | 55.98 |
| TENT+ | 33.96 | 36.66 | 35.75 | 33.70 | 33.33 | 47.73 | 53.22 | 52.16 | 44.79 | 60.62 | 68.91 | 35.60 | 58.81 | 61.82 | 56.23 |
| HLR (ours) | 38.39 | 41.11 | 40.28 | 38.25 | 38.18 | 51.63 | 55.55 | 55.45 | 48.96 | 62.19 | 68.17 | 49.47 | 60.34 | 62.51 | 57.42 |
| SLR (ours) | **39.51** | **42.09** | **41.58** | **39.35** | **39.02** | **52.67** | **55.80** | **55.92** | **49.64** | **62.62** | **68.47** | **50.27** | **60.80** | **63.01** | **57.80** |
| Epoch 5 | | | | | | | | | | | | | | | |
| TENT | 16.04 | 23.22 | 25.85 | 19.05 | 17.40 | 49.02 | 52.78 | 52.72 | 34.31 | 61.19 | 68.54 | 1.26 | 59.26 | 62.15 | 56.17 |
| TENT+ | 33.97 | 37.95 | 36.93 | 32.69 | 33.36 | 51.42 | 54.33 | 54.55 | 45.80 | 62.09 | 69.03 | 24.08 | 60.36 | 63.10 | 57.21 |
| HLR (ours) | 41.37 | **44.04** | 43.68 | **41.74** | **41.09** | 54.26 | 56.43 | 57.03 | 50.81 | 63.05 | 68.29 | 50.98 | 61.15 | 63.08 | 58.13 |
| SLR (ours) | **41.52** | 42.90 | **44.07** | 41.69 | 40.78 | **54.76** | **56.59** | **57.35** | **51.01** | **63.53** | **68.72** | 50.65 | **61.49** | **63.46** | **58.32** |
| Groundtruth | 55.68 | 58.10 | 61.27 | 55.84 | 55.08 | 65.83 | 67.22 | 67.56 | 62.60 | 72.49 | 76.97 | 65.04 | 70.86 | 72.51 | 68.56 |

Table 2: SSIM and SLR-adapted ResNet50 accuracy without and with input transformation (IT).

| Corruption | Gauss | Shot | Impulse | Defocus | Glass | Motion | Zoom | Snow | Frost | Fog | Bright | Contrast | Elastic | Pixel | JPEG |
|---|---|---|---|---|---|---|---|---|---|---|---|---|---|---|---|
| SSIM | 0.123 | 0.147 | 0.135 | **0.623** | **0.648** | **0.622** | **0.676** | 0.517 | 0.575 | 0.619 | 0.653 | 0.545 | **0.625** | **0.786** | **0.800** |
| SSIM+IT | **0.173** | **0.188** | **0.347** | 0.605 | 0.638 | 0.603 | 0.670 | **0.580** | **0.628** | **0.626** | **0.676** | **0.765** | 0.616 | 0.776 | 0.795 |
| SLR | 41.59 | 43.49 | 43.90 | 41.70 | **41.10** | 54.86 | 56.39 | 57.47 | 50.90 | 63.51 | 68.70 | 51.06 | **61.36** | 63.39 | **58.35** |
| SLR+IT | **43.09** | **44.39** | **64.05** | **41.98** | 40.99 | **55.73** | **56.75** | **58.56** | **51.68** | **63.64** | **68.85** | **55.01** | 61.32 | **63.59** | 58.24 |

statistics are still updated in these layers. Please refer Sec. A.2 for more details regarding which layers are frozen in different networks.

Furthermore, we prepend a trainable input transformation module $d$ (cf. Sec. 3.1) to the network to partially counteract the data-shift. Note that the parameters of this module discussed in Sec. 3.1 are trainable and subject to optimization. This module is initialized to operate as an identity function prior to adaptation on a target distribution by choosing $\tau = 1$, $\gamma = 1$, and $\beta = 0$. We adapt the parameters of this module along with the channel-wise affine transformations and normalization statistics in an end-to-end fashion, solely using our proposed loss function along with the optimization details mentioned above. The architecture of this module is discussed in Sec. A.1.

Since $L_{\mathrm{div}}$ is independent of $L_{\mathrm{conf}}$, we also propose to combine $L_{\mathrm{div}}$ with TENT, i.e. $L = L_{\mathrm{div}} + L_{ent}$. We denote this as TENT+ and also set $\kappa = 0.9$ here. Note that TENT optimizes all channel-wise affine parameters in the network (since entropy is saturating and does not cause logit explosion). For a fair comparison to our method, we also freeze the top layers of the networks in TENT+. We show that adding $L_{\mathrm{div}}$ and freezing top layers significantly improves the networks performance over TENT. Note that SHOT [11] is the combination of TENT, batch-level diversity regularizer, and their pseudo labeling strategy. TENT+ can be seen as a variant of SHOT but without their pseudo labeling technique. Please refer to Sec. A.4 for the test-time adaptation of pretrained models with SHOT.

Note that each corruption and each severity in ImageNet-C is treated as a different target distribution and in all settings we reset model parameters to their pretrained values before every adaptation. We run our experiments for three times with different random seeds (2020, 2021, 2022) in PyTorch and report the average accuracies.

## 5 Results

**Evaluation on ImageNet-C** We adapt different models on the ImageNet-C benchmark using TENT, TENT+, and both *hard likelihood ratio* (HLR) and *soft likelihood ratio* (SLR) losses. Figure 2 (top row) depicts the mean corruption accuracy (mCA%) of each model computed across all the corruptions and severity levels. It can be observed that TENT+ improves over TENT, showcasing the importance of a diversity regularizer $L_{\mathrm{div}}$. Importantly, our methods HLR and SLR outperform TENT and TENT+ across DenseNet121, MobileNetV2, ResNet50, ResNeXt50 and perform comparable with TENT+ on robust ResNet50-DeepAugment+Augmix model. This shows that the mCA% of robust DeepAugment+Augmix model can be further increased from 58% (before adaptation) to 68.6% using test-time adaptation techniques. Here, the average of mCA obtained from three different

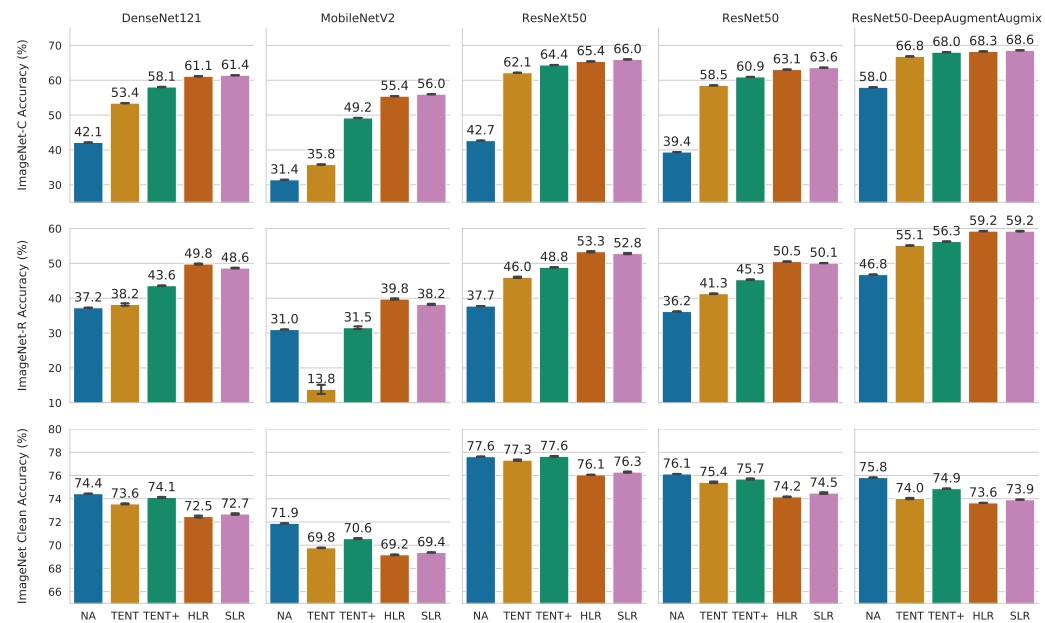

Figure 2: Test-time adaptation results on (top row) ImageNet-C, averaged across all 15 corruptions and severities, (middle row) ImageNet-R, (bottom row) clean ImageNet. NA refers to "No Adaptation".

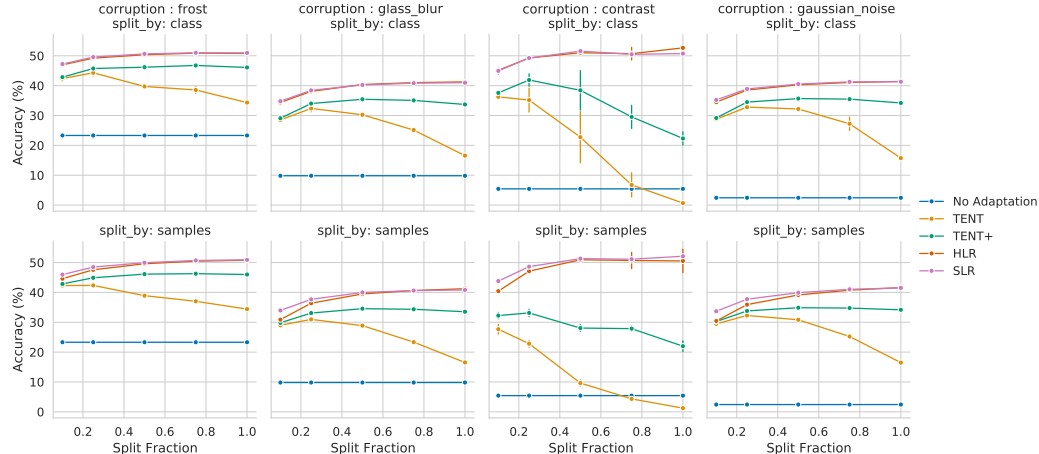

Figure 3: Test-time adaptation of ResNet50 using (top row) a subset of classes, and (bottom row) a subset of samples per class on 4 different corruptions at severity 5. Accuracy is computed based on the evaluation of adapted model on the entire target data. Note that error bars are smaller to visualize.

random seeds are depicted along with the error bars. These smaller error bars represent that the test-time adaptation results are not sensitive to the choice of random seed.

We also illustrate the performance of ResNet50 on the highest severity level across all 15 test corruptions of ImageNet-C in Table 1. Here, the adaptation results after epoch 1 and 5 are reported. It can be seen that a single epoch of test-time adaptation improves the performance significantly and makes minor improvements until epoch 5. TENT adaptation for more than one epoch result in reduced performance and TENT with $L_{\text{div}}$ (TENT+) prevents this behavior. We note that both HLR and SLR clearly and consistently outperform TENT and TENT+ on the ResNet50. We also compare our results with the hard pseudo-labels (PL) objective and also with an oracle setting where the groundtruth labels of the target data are used for adapting the model in a supervised manner (GT). Note that this oracle setting is not of practical importance but illustrates the empirical upper bound on fully test-time adaptation performance under the chosen modulation parametrization. The reported numbers in the table are the average of three random seeds.

**ImageNet-R** We evaluate different adapted models on ImageNet-R and depict the results in Figure 2 (middle row). Results show that our methods significantly improve performance of all the models, including the model pretrained with DeepAugment+Augmix. Moreover, both HLR and SLR clearly outperform TENT and TENT+.

**Evaluation with data subsets** In the above experiments, the model is evaluated on the same data that is also used for the test-time adaptation. Here, we test model generalization by adapting on a subset of target data and evaluate the performance on the whole dataset, which also includes unseen data that is not used for adaptation. We conduct two case studies: (i) adapt on the data from a subset of ImageNet classes and evaluate the performance on the data from all the classes. (ii) Adapt only on a subset of data from each class and test on all seen and unseen samples from the whole dataset.

Figure 3 illustrates generalization of a ResNet50 adapted on different proportions of the data across different corruptions, both in terms of classes and samples. We observe that adapting a model on a small subset of samples and classes is sufficient to achieve reasonable accuracy on the whole target data. This suggests that the adaptation actually learns to compensate the data shift rather than overfitting to the adapted samples or classes. The performance of TENT decreases as the number of classes/samples increases, because $L_{ent}$ can converge to trivial collapsed solutions and more data corresponds to more updates steps during adaptation. Adding $L_{\text{div}}$ such as in TENT+ stabilizes the adaptation process and reduces this issues. Reported are the average of random seeds with error bars.

**Input transformation** We investigate whether the input transformation (IT) module, trained end-to-end with a ResNet50 and SLR loss on data of the respective distortion *without* seeing any source (undistorted) data, can partially undo certain domain shifts of ImageNet-C and also increase accuracy on corrupted data. We measure domain shift via the structural similarity index measure (SSIM) [44] between the clean image (unseen by the model) and its distorted version/the output of IT on the distorted version. Table 2 shows that IT increases the SSIM considerably on certain distortions such as Impulse, Contrast, Snow, and Frost. IT increases SSIM also for other types of noise distortions, while it slightly reduces SSIM for the blur distortions, Elastic, Pixelate, and JPEG. When combined with SLR, IT considerably increases accuracy on distortions for which also SSIM increased significantly (for instance +20 percent points on Impulse, +4 percent points on Contrast) and never reduces accuracy by more than 0.11 percent points. We provide illustrations of effect of IT in the appendix.

**Clean images** As a sanity check, we investigate the effect of test-time adaptation when target data comes from the same distribution as training data. For this, we adapt pretrained models on clean validation data of ImageNet. The results in Figure 2 (bottom row) depict that the performance of SLR/HLR adapted models drops by $1.5$ to $2.5$ percent points compared to the pretrained model. We attribute this drop to self-supervision being less reliable than the original full supervision on in-distribution training data. The drop is smaller for TENT and TENT+, presumably because predictions on in-distribution target data are typically highly confident such that there is little gradient and thus little change to the pretrained networks by TENT. In summary, while self-supervision by confidence maximization is a powerful method for adaptation to domain shift, the observed drop when adapting to data from the source domain indicates that there is "no free lunch" in test-time adaptation.

# 6   Conclusion

We propose a method to improve corruption robustness and domain adaptation of models in a fully test-time adaptation setting. Unlike entropy minimization, our proposed loss functions provide non-vanishing gradients for high confident predictions and thus attribute to improved adaptation in a self-supervised manner. We also show that additional diversity regularization on the model predictions is crucial to prevent trivial solutions and stabilize the adaptation process. Lastly, we introduce a trainable input transformation module that partially refines the corrupted samples to support the adaptation. We show that our method improves corruption robustness on ImageNet-C and domain adaptation to ImageNet-R on different ImageNet models. We also show that adaptation on a small fraction of data and classes is sufficient to generalize to unseen target data and classes.

**Ethical and Societal Impact**   Our non-saturating loss increases accuracy but might result in over-confident predictions, which can cause harm in safety-critical downstream applications when not properly calibrated. At the same time, self-supervised confidence maximization might amplify bias in pretrained models. We hope that the diversity regularizer in the loss partially compensates this issue.

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
