# OpenReview forum: "Test-Time Adaptation to Distribution Shifts by Confidence Maximization and Input Transformation"
_NeurIPS.cc/2021/Conference — NeurIPS 2021 Submitted_

### Official Review · Reviewer_mx5v · 2021-07-16

**Rating:** 6
**Confidence:** 4

**Summary:**

Fully test-time adaptation is a promising approach to handling data shift and making the DNNs more robust. This paper presents a new method for fully test-time adaptation. Specifically, they propose a non-saturating surrogate of entropy and combine it with the diversity regularizer. They empirically demonstrate the efficacy of the proposed method, comparing it against a recently proposed test-time adaptation method, Tent, using various backbone networks.

**Limitations And Societal Impact:**

The paper mentions a little about the difficulty of using the method in safety-critical downstream tasks and describes the diversity regularizer possibly compensates the issue. However, it is unclear to me how the proposed regularizer help to migrate the issue. I feel this is the most critical issue of fully test-time adaptation, and it is worth discussing more.

Besides that, I can not find any statements about limitations. For example, it can not directly be used for the regression, as the non-saturating modification assumes the discrete prediction (although Tent has the same issue). Would you please discuss more about such a limitation of this work?

**Main Review:**

---
Overall, the paper is very clearly written, and it was a fan to read. The proposed method might not be a revolutionary one, a combination of several (known) techniques, yet it is well described, and the experimental results are extensive enough to show benefits. They also provide results on various backbone networks, which is informative for future researches. However, several details regarding the motivation and implementation are unclear for me, making me hesitate to vote for full acceptance of the paper.

---
**Comments/Questions**

1. The motivation behind the proposal of a non-saturating objective is that information about more confident samples should be affected more on test-time adaptation. However, I do not fully agree with this point, as it does not affect the training just because the model parameters are already suitable for the data point.
2. If I correctly understand the paper, there are two main differences between the Tent and the proposed method: (1) loss function and (2) tuned parameter. However, it is not clear to me why do we need to change the latter one. Can you discuss more that? Specifically, what is the potential problem of transforming only the parameter of BN as with the Tent?
3. Figure 2 suggests that the proposed method often negatively affects the performance on the clean dataset, compared to the Tent and Tent+. Why does it happen? Besides, it is not clear to me how exactly the performance on the clean dataset is measured. More specifically, is the model adapted on the clean dataset or adapted on the corrupted dataset?
4. In section 4, the authors say that the experiment uses cosine decay scheduling. However, looking at the code, this doesn't seem to apply to Tent.  It seems that the comparison is unfair due to this.
5. Learning five epochs during the test seems a bit strange to me. This is because the deployed model usually needs to make correct predictions at that moment; there is no point in going back in time and correcting the predictions.

**Time Spent Reviewing:**

3 hours

---

> ### Author Response · Authors · 2021-08-10
> **Thank you for raising interesting questions. It further refines the understanding of our approach.**
>
> **1) Motivation behind the proposal of a non-saturating objective:**
> The motivation for confidence maximization-type adaptation is that the decision boundary should be "far" from actual datapoints and the prediction confidence acts as a proxy for this distance. On the one hand, the less confident the prediction on a datapoint, the closer is the datapoint to the decision boundary and the stronger the datapoint should be pushed  away from the boundary (large gradient). However, on the other hand, the lower the confidence, the less certain it becomes on which side of the decision boundary a datapoint should lie (what we call the "low confidence in self-supervision"), and thus the direction of the gradient becomes ambiguous. As evidenced by our empirical results, increasing the influence of high confidence predictions on the gradient as with our proposed losses improves performance. We argue that this is the case because it leads to better self-supervision (better gradient direction when averaged over the batch) than for the entropy loss, even though those highly confident predictions are already far from the decision boundary.
>
>  **2) What is the potential problem of transforming only the parameter of BN as with the Tent?**
>  We also adapt the BN statistics and optimize only the channel-wise affine parameters of the network on the target distribution, similar to TENT (mentioned in Line 249). In other words, we optimize the same parameters that TENT optimize. Differences to TENT comes with i) freezing top layers of the networks and ii) input transformation module. We freeze network top layers to avoid logit explosion when using our proposed likelihood ratio loss (please refer Line 225-231 in the main paper for details) and using an additional input transformation module gives the model a larger adaptation capability compared to adapting only the parameters of BN.
>
> **3) How the performance on the clean dataset is measured?**
> Performance on the clean dataset is measured by adapting the ImageNet pretrained network on the clean dataset using test-time adaptation objectives (TENT/TENT+/HLR/SLR). When measuring the clean data performance, no corrupted dataset was used. Here, the network is solely adapted on the clean dataset from the ImageNet pretrained values for 5 epochs and later evaluated on the same. We will make this detail clear in our final version.
>
> **Why proposed method negatively affects the performance on the clean dataset?:**
> Model predictions on in-distribution target data (clean dataset) are typically highly confident. TENT and TENT+ provide smaller gradients on such highly confident predictions and thus make little changes to the pretrained networks. On the other hand, our proposed likelihood ratio provide relatively larger gradients than TENT/TENT+ even on these highly confident predictions. Hence, the highly confident wrongly classified samples acts as data with noisy labels, which have influence during adaptation and affect the performance.
>
> **4) Cosine decay scheduling on TENT**
>  Thanks for the suggestion. Below table presents ResNet50 results for all corruptions at severity 5 adapted for 5 epochs with cosine decay scheduling on TENT and TENT+.   Reported are the mean accuracy(\%) across three random seeds (2020/2021/2022). Note that the error bars here are in a smaller scale or negligible as similar to the error bars reported in our appendix. TENT and TENT+ obtain improved results with cosine decay scheduler, however SLR still outperforms them.
>
>  ||Gauss|Shot|Impulse|Defocus|Glass|Motion|Zoom|Snow|Frost|Fog|Bright|Contrast|Elastic|Pixel|JPEG|
> |:---------:|:---------:|:---------:|:---------:|:---------:|:---------:|:---------:|:---------:|:---------:|:---------:|:---------:|:---------:|:---------:|:---------:|:---------:|:---------:|
> |TENT(no cosine decay)|16.04|23.22|25.85|19.05|17.40|49.02|52.78|52.72|34.31|61.19|68.54|1.26|59.26|62.15|56.17|
> |TENT(with cosine decay)|30.64|33.80|34.72|30.13|29.05|49.08|53.63|52.86|38.47|61.13|68.81|10.72|59.25|62.15|56.44|
> |TENT+(no cosine decay)|33.97|37.95|36.93|32.69|33.36|51.42|54.33|54.55|45.80|62.09|69.03|24.08|60.36|63.10|57.21|
> |TENT+(with cosine decay)|35.19|38.12|37.43|34.82|34.95|50.33|54.24|53.88|46.28|61.50|**69.07**|29.87|60.01|62.61|57.09|
> |SLR(cosine decay)|**41.52**|**42.90**|**44.07**|**41.69**|**40.78**|**54.76**|**56.59**|**57.35**|**51.01**|**63.53**|68.72|**50.65**|**61.49**|**63.46**|**58.32**|
>
> **5) Deployed model needs to make correct predictions at that moment; not going back in time and correcting the predictions.**
>  We show in Table 1 that, adapting the model for 5 epochs brings only minor improvements over the model that is adapted only for 1 epoch. However, we agree with the reviewer that it is imperative to make correct predictions at that moment in certain critical real world applications. Following that, we conducted an experimental setting where model predicts on a batch directly after it adapts on the batch i.e., making predictions at that moment. Below table presents the results of different adaptation objectives with model predictions on a batch directly after adapting on the batch. Results are obtained with ResNet50 for all corruptions at severity 5 and reported mean accuracy(\%) across three random seeds. These numbers are obtained during the 1st epoch of adaptation but not after the 1st epoch of adaptation. We see that SLR still outperforms all the other methods in this setting.
>
> ||Gauss|Shot|Impulse|Defocus|Glass|Motion|Zoom|Snow|Frost|Fog|Bright|Contrast|Elastic|Pixel|JPEG|
> |:---------:|:---------:|:---------:|:---------:|:---------:|:---------:|:---------:|:---------:|:---------:|:---------:|:---------:|:---------:|:---------:|:---------:|:---------:|:---------:|
> |TENT (no cosinedecay)|28.69|30.98|30.41|29.09|28.01|42.46|50.43|48.15|42.03|58.48|68.26|26.94|55.66|59.52|53.52|
> |TENT (cosine decay)|28.65|30.93|30.34|29.02|27.97|42.41|50.41|48.11|42.04|58.45|68.25|27.08|55.65|59.51|53.52|
> |TENT+ (no cosine decay)|29.21|31.52|30.57|29.19|28.59|42.55|50.47|48.17|42.51|58.51|68.29|31.24|55.76|59.54|53.60|
> |TENT+ (cosine decay)|29.18|31.49|30.51|29.17|28.55|42.50|50.45|48.13|42.50|58.49|68.29|31.23|55.73|59.52|53.59|
> |HLR (cosine decay)|33.11|35.99|35.15|33.34|33.35|46.61|52.97|51.55|45.67|60.10|68.04|42.97|57.97|60.40|55.07|
> |SLR (cosine decay)|**35.04**|**37.86**|**37.08**|**35.22**|**35.12**|**48.50**|**53.74**|**52.89**|**46.72**|**60.75**|**68.32**|**44.91**|**58.86**|**61.21**|**55.90**|
>
> **6) How the proposed diversity regularizer help?:**
> Confidence maximization is prone to converge to collapsed solutions that always predict a single or a few classes. It can thus exhibit a learning dynamic that overly favors classes that are over-represented in the target domain data. The diversity regularizer is a way of clamping class frequencies close to some target class frequency and can thus prevents class collapse. In general, we agree there is a need for further research on preventing bias in unsupervised domain adaptation and test-time adaptation.
>
> **7) Statements about limitations?:**
> Thanks for the suggestion, we will include that our method can not directly be used for the regression as a limitation. We summarize other limitations discussed in the paper in Checklist 1b.

---

> > ### Comment · Reviewer_mx5v · 2021-08-24
> > **Additional questions and comments**
> >
> > Thanks for the response, which addresses some of my concerns. However, I have several follow-up questions and comments.
> >
> > **(1) Test-time adaptation and SFDA are different.**  In the current manuscript, test-time adaptation seems to have the same meaning as source-free adaptation. However, these words have slightly different meanings since the former assumes the online data, yet the latter takes offline data for unlabeled data. The distinction is also clearly written in the TENT paper. The paper should at least discuss more on this point and clarify the difference in position with prior works. In addition, I think that the experimental parts should distinguish these two setups for clarification.
> >
> > The same issue inherits to the experimental results. My main concern is that the current paper is only compared with the online-adaptation method (TENT and its variants) and not with the SFDA method. At the same time, the setting is more suitable for calling source-free domain adaptation since the experimental setting assumes that we can access the offline dataset. Comparison with SFDA methods strengthens the paper.
> >
> > **(2)The meaning of cosine decay scheduler w/ one epoch is ambiguous**. Thank you for providing additional experimental results. I believe it possibly strengthen the paper. However, I'm not sure of the meaning of schedule when optimizing it one epoch (online setting).
> > Did you schedule the learning rate based on the batch number (I think it is usually updated based on the epoch number)? How often update it, and how did you determine it? Since the hyper-parameter selection in online adaptation is severe, the selection method is important.
> > Similar to the above question, how can we determine the max iteration, which also determines the scheduling of LR?
> >
> >
> > Overall, I am still not confused about the paper's position (especially regarding the offline adaptation vs. online adaptation) and related experimental issues. I, therefore, would like to keep my score.

---

> > > ### Author Response · Authors · 2021-08-26
> > > **Thank you for raising interesting comments**
> > >
> > > (1) **Test-time adaptation and SFDA:** In the Related Work section of our paper, we have discussed the differences between test-time adaptation and source-free adaptation methods and the rationale behind categorizing our work into test-time adaptation. As noted, prior works on source free adaptation use generative models [29, 30, 31, 32], require more computation (several thousand epochs to adapt to the target data) [30, 32], and also alter training [11]. On the other hand, our work falls in the research line of test-time adaptation [10, 33, 34, 35, 1] that aim to improve the model robustness against common corruptions with less computational overhead to facilitate online adaptation. Our work replaces the entropy minimization from TENT  with the proposed losses and thus allows the test-time adaptation on arbitrary pretrained models (without altering training) with less computational overhead. Such setting enables our approach to be deployed more generally to facilitate both offline and online adaptation, same as TENT paper which benchmark the adaptation with both offline and online updates. Results posted in the above response and this response (see below) demonstrate that our approach outperforms TENT and its variants also during the online adaptation. We thank the reviewer for this suggestion. We will elaborate this discussion and explain the differences with prior works in our final version. We will also distinguish offline and online adaptation settings in our experimental setup and provide the results for both the settings in our final version.
> > > Since our approach focus on adaptation with less computation overhead, no generative modeling and no alteration of training, we primarily compare with TENT and its variants in the main paper. We also shown comparison of our work with one of the popular source-free adaptation technique SHOT [11] in the appendix A.4.
> > >
> > >
> > > (2) **cosine decay scheduler w/ one epoch:** For the cosine decay schedule, we schedule the learning rate based upon the batch number (i.e., update step number). We set the maximum update steps to be 4000. However, a single epoch with batch size 64 on a corruption dataset of ImageNet-C counts for only 782 update steps (number of samples 50000/batch size 64) for one epoch. Though we set the maximum update steps to 4000, the reported numbers for the online adaptation are obtained only during the first 782 update steps (1 epoch). We agree with the reviewer that the hyper-parameter selection for the learning rate schedule during online adaptation is not straightforward. We thank the reviewer for pointing us this detail. For this reason, we conduct the test-time adaptation with HLR and SLR with constant learning rate (no learning rate scheduler) to facilitate the online updates. Below we present the results of HLR and SLR without cosine decay scheduler during online adaptation. Note that the results with cosine decay scheduler (previous response) and without cosine decay scheduler (below) are similar during the online adaptation. This is because we set the maximum update steps to a larger number than the update steps in one epoch when cosine decay scheduler was enabled. We will include the results on online adaptation in our final version.
> > >
> > > ||Gauss|Shot|Impulse|Defocus|Glass|Motion|Zoom|Snow|Frost|Fog|Bright|Contrast|Elastic|Pixel|JPEG|
> > > |:---------:|:---------:|:---------:|:---------:|:---------:|:---------:|:---------:|:---------:|:---------:|:---------:|:---------:|:---------:|:---------:|:---------:|:---------:|:---------:|
> > > |TENT|28.69|30.98|30.41|29.09|28.01|42.46|50.43|48.15|42.03|58.48|68.26|26.94|55.66|59.52|53.52|
> > > |TENT+ |29.21|31.52|30.57|29.19|28.59|42.55|50.47|48.17|42.51|58.51|68.29|31.24|55.76|59.54|53.60|
> > > |HLR|33.04|35.88|35.24|33.23|33.31|46.69|52.95|51.50|45.66|60.05|67.94|42.75|57.95|60.37|55.15|
> > > |SLR|**35.01**|**37.88**|**37.02**|**35.20**|**35.10**|**48.51**|**53.75**|**52.91**|**46.72**|**60.77**|**68.35**|**44.99**|**58.80**|**61.20**|**55.89**|
> > >
> > > Similar to the previous response, these results are obtained with ResNet50 for all corruptions at severity 5 and reported mean accuracy(%) across three random seeds.

---

### Official Review · Reviewer_ab5L · 2021-07-17

**Rating:** 6
**Confidence:** 4

**Summary:**

Focuses on test-time adaptation to distribution shifts, and proposes: i) a likelihood-ratio based loss with non-saturating gradients, ii) a diversity regularizer based on batch-wise entropy maximization, and iii) a learned input transformation module to undo the effect of distribution shift. Results are presented on standard test-time adaptation benchmarks.

**Limitations And Societal Impact:**

Authors adequately addressed the limitations and potential negative societal impact.

**Main Review:**

Originality

– This work extends prior work (TENT) to optimize only batch norm parameters in addition to another “input transformation” module, to minimize a different loss function.

– The originality of the stated technical contributions is somewhat limited:

i) As the authors acknowledge (L215), Yao et al. [40] were the first to propose the negative log likelihood-ratio loss

ii) the diversity regularizer has also been proposed in prior work [11, 12]. The authors claim that they extend this to multiple batches, but this has also been done in Li et al. [A], which regularizes diversity by maximizing entropy over a history of model predictions over batches stored in a queue.

– The shortcomings of the entropy minimization loss have been studied in prior work in domain adaptation: eg. [B] which proposes a new “max squares loss” and [C] which applies a Charbonnier penalty to entropy minimization. This work i) misses citing these highly related works, ii) clarifying what the contributions over these existing works is iii) empirically showing how the proposed loss compares against these prior works.

[A] Li, Bo, et al. "Rethinking distributional matching based domain adaptation." arXiv preprint arXiv:2006.13352 (2020).

[B] Chen, Minghao, Hongyang Xue, and Deng Cai. "Domain adaptation for semantic segmentation with maximum squares loss." Proceedings of the IEEE/CVF International Conference on Computer Vision. 2019.

[C] Yang, Yanchao, and Stefano Soatto. "Fda: Fourier domain adaptation for semantic segmentation." Proceedings of the IEEE/CVF Conference on Computer Vision and Pattern Recognition. 2020.

Quality

– The improvements over TENT on test-time adaptation are convincing, and the authors do well to benchmark multiple architectures.

– The paper lacks a clear ablation study -- only ITM appears to be ablated. What is the effect of removing L_{DIV} / L_{SLR}? Or optimizing L_{DIV} only over the current batch rather than as a moving average?

– It’s also not entirely convincing that ITM helps – Table 2 shows that SSIM (SLR) > SSIM (SLR+IT) in 7/15 settings (though admittedly by narrower margins as the authors mention). In my opinion, this is inadequate evidence to claim that “ITM partially undoes the distribution shift”. The authors say that “illustrations of the effect of IT” is included in appendix, but I was unable to find this experiment.

Clarity

– The design of input transformation module is not clearly motivated. It appears to be a channel-wise learned affine transformation applied to a convex combination of the input and its (nonlinear) transformed version – while the first part seems identical to a batch-normalization layer, what is the motivation/intuition behind the second part? Have the authors experimented with alternatives?

– L162 states “assuming knowledge of the class distribution .. on the target domain”, but does not clarify how this is estimated / approximated. Lacking access to labels on the target, how is p_{D’}(y) computed? Or is it assumed to be the uniform distribution?

– Minor: Why is there a separate “pseudo labels” row in Table 1? Is it not identical to the previous “no adaptation” row by definition?

– In Figure 2, HLR outperforms SLR on ImageNet-C and clean ImageNet, but the opposite happens on ImageNet-R. Have the authors
investigated the reason for this? When is it more appropriate to use one over the other?

– Minor: Why is SHOT not directly included as a point of comparison in say Table 1, but rather in the appendix?

Significance

– The paper studies an interesting and practical setting of test-time adaptation, and the proposed approach appears to improve upon the existing state-of-the-art (TENT) on standard benchmarks.

Overall

The paper studies the important and practical setting of test-time adaptation, and extends prior work (TENT) via a new loss function with non-saturating gradients and a new input transformation module, which lead to improved performance. The paper is mostly well-written and easy to follow. I have concerns primarily about the originality of the paper’s stated contributions (primarily, not citing / comparing to highly relevant work in DA studying the shortcomings of entropy minimization) and the quality of its experimental section (primarily ablations), and so believe that in its current form it is below the bar for acceptance.


---post-rebuttal---

I have read the author response and other reviews. While I still believe the technical novelty is slightly limited, I think the paper is well-written and the rebuttal addresses most of my primary concerns, and recommend marginal acceptance.

**Time Spent Reviewing:**

3

---

> ### Author Response · Authors · 2021-08-10
> **Thank you for pointing the related works. The additional comparisons support our approach.**
>
> **1) Comparison to previous works:**
> We thank the reviewer for pointing out the additional references. We will include and cite them in our final version. Li et al. [A] use a moving average to estimate the entropy of the unconditional class distribution. However, to estimate the gradient of the entropy, the source data is used, c.f. equation (3) in [A]. Our method uses a moving average as well, but in contrast to [A], our method works source free, i.e. we do not need access to the source data during test time adaptation. The gradient is estimated using only the test data. We will stress this difference in our final version and see it as one of the contributions in our Section 3.2.1.
>
> Both the Max Squares loss [B] and the Charbonnier penalty applied to entropy minimization [C] have gradients of the loss w.r.t. the logit that go to 0 for high confidence predictions (confidences greater than 0.95). This is not the case for our proposed non-saturating losses, which we see as a contribution over these works. This vanishing gradient leads to the disadvantages discussed in our paper, i.e. high confidence predictions have less contribution on the test-time adaptation than lower confident predictions. Please refer the first response to Reviewer1 behind the motivation of our proposed loss. Additional empirical results on ImageNet-C shown below support that SLR outperforms the losses from [B] and [C], even when tuning the Charbonnier penalty's hyperparamter eta carefully.
>
> ||Gauss|Shot|Impulse|Defocus|Glass|Motion|Zoom|Snow|Frost|Fog|Bright|Contrast|Elastic|Pixel|JPEG|
> |:---------:|:---------:|:---------:|:---------:|:---------:|:---------:|:---------:|:---------:|:---------:|:---------:|:---------:|:---------:|:---------:|:---------:|:---------:|:---------:|
> |Max_Square|29.08|31.13|30.66|29.93|28.35|44.22|52.12|49.96|44.21|59.96|68.75|34.85|57.51|61.07|55.03|
> |Max_Square+$L_{div}$|28.28|31.30|29.33|29.21|28.09|43.37|51.59|49.22|43.73|59.58|68.64|40.66|57.31|60.65|54.60|
> |Charbonnier_Penalty(eta=0.3)|33.46|36.34|36.58|32.40|31.44|50.33|54.50|53.85|40.38|61.71|69.00|13.63|59.93|62.58|57.03|
> |Charbonnier_Penalty+$L_{div}$ (eta=0.3)|36.42|39.20|38.50|36.40|36.10|51.08|54.86|54.47|47.43|61.93|**69.15**|36.59|60.42|62.99|57.48|
> |SLR+$L_{div}$ (ours)|**39.51**|**42.09**|**41.58**|**39.35**|**39.02**|**52.67**|**55.80**|**55.92**|**49.64**|**62.62**|68.47|**50.27**|**60.80**|**63.01**|**57.80**|
>
> Here, different eta values [0.1, 0.3, 0.75, 1.0, 1.75, 2.0]  are explored for Charbonnier penalty and found eta=0.3 performs better. Reported values are mean accuracy(\%) over three random seeds (2020/2021/2022) of ResNet50 at severity level 5. Note that the error bars here are in a smaller scale or negligible as similar to the error bars reported in our appendix.
>
> **2) Ablation studies on  $L_{div}$ / $L_{slr}$:**
> Below table presents adaptation accuracy(\%) after (epoch1 / epoch5) on validation corruptions of ImageNet-C at severity 5 on ResNet50 across three random seeds (mean value is presented here). $L_{slr}$ + batch $L_{div}$ act on a batch of samples, hence the number of classes has to be smaller than the batch size. However, it is critical to include the history of the previous batches when handling with larger number of classes and smaller batch size (e.g. ImageNet 1000 classes with batch size 64), hence extended the batch $L_{div}$ to moving average $L_{div}$. It is evident that only the full $L_{slr}$ + running $L_{div}$  performs well across all corruptions and all ablations fail catastrophically on the Gaussian blur corruption.
>
> ||Speckle_noise|Gaussian_blur|Spatter|Saturate|
> |:---------:|:---------:|:---------:|:---------:|:---------:|
> |batch $L_{div}$|0.17/0.11|0.18/0.10|0.17/0.12|0.25/0.15|
> |moving $L_{div}$|0.27/0.12|2.47/1.12|4.66/0.45|1.76/0.20|
> |$L_{slr}$|49.38/23.87|4.26/1.36|60.03/55.66|64.81/65.29|
> |$L_{slr}$ + batch $L_{div}$|0.18/0.14|0.36/0.24|0.25/0.13|0.26/0.14|
> |$L_{slr}$ + moving $L_{div}$|**53.12/54.73**|**33.87/36.10**|**60.51/61.94**|**64.84/65.41**|
>
> **3) Mixed results from input transformation module:**
> We use SSIM to measure the domain shift, and however we suggest to evaluate the benefit of input transformation (IT) rather on the accuracy of the adapted network (bottom two rows of Table 2). Here, IT considerably increases accuracy on some distortions (for instance +20.0 percent points on Impulse,  +4.0 percent points on Contrast) and never reduces accuracy by more than 0.11 percent point. It improves performance on 11 out of the 15 distortion types. Thus the possible performance gains clearly outweigh the possible performance drops, and we would not call it a "mixed" result.
>
> We sincerely apologize for the mistake that “illustrations of the effect of IT” is missing from the appendix. We will include the illustration in our final version.
>
> **4) Design of input transformation:**
> The specific choice of the second part ($r_\psi$) is illustrated in Figure A.1. This choice is motivated by the general observation that deep networks excel at image-to-image translation tasks (and undoing appearance shifts in images is such a task). We do not claim that the specific choice of the network architecture is optimal; alternative choices such as ones based on Transformers or with other types of long-range interactions might be preferable. We leave this investigation to future work.
>
> **5) Knowledge of the class distribution:**
> In principle, if additional information about the target label distribution is available, it could be used with the proposed method. This could be the case in some applications, e.g. in medical data it might be known that there is a large class imbalance. In general this information is not available, and here we assume a uniform distribution $p_{D’}(y)$.
>
> **6) Which is better? HLR or SLR?**
> In most of our experiments, SLR outperforms HLR (albeit the difference is generally relatively small). In particular, the bottom row of Figure 3 indicates that SLR is preferable in the "low-data regime". We thus consider SLR the default choice. Why HLR performs better than SLR for some models on ImageNet-R is an interesting question, which we cannot conclusively answer.
>
> **7) Why a separate “pseudo labels” row in Table 1? Is it not identical to “no adaptation”?**
> "Pseudo Labels" corresponds to adaptation based on the loss $L_{pl}$ (Line 189). It is not guaranteed to have the same accuracy as "no adaptation" in general, albeit we observe this in our experiments.
>
> **8) Why is SHOT not in Table 1, but rather in the appendix?**
> Table1 comprises of publicly available ImageNet pretrained models from torchvision. As mentioned in Line 106, SHOT modifies the model from the standard setting by adopting weight normalization at the fully connected classifier layer during training to facilitate their pseudo labeling technique. Hence, SHOT is not readily applicable to arbitrary pretrained models as in Table1. In the appendix, we follow the instructions from SHOT and modify ResNet50 accordingly to enable a fair comparison.

---

> > ### Comment · Reviewer_ab5L · 2021-08-23
> > **Follow-up questions**
> >
> > Thanks for the clarifications and additional experiments which address some of my concerns. I have a few follow-up questions:
> >
> > 1) The authors are correct that Li et al. appears to use source data for computing gradients for their diversity loss, but prior work has also optimized this loss using target data (eg. see Eq. 3 in Prabhu et al.). That said, I think it is reasonable to consider this as contemporary work – it would nevertheless be good to cite and discuss similarities for completeness.
> >
> > [A] Prabhu, Viraj, et al. "SENTRY: Selective Entropy Optimization via Committee Consistency for Unsupervised Domain Adaptation." arXiv 2020.
> >
> > 2) I appreciate the authors' efforts in including conceptual and empirical comparisons with the MaxSquares and Charbonnier penalty approaches from prior work, and the presented results look promising. For clarity, what exactly do the presented results correspond to, considering that both prior works are not designed for the test-time adaptation setting? Are they ablations of the proposed method but with MaxSquares / Charbonnier penalty instead of the SLR/HLR loss? Do they also optimize only batch-norm parameters? I would consider training only BN parameters + using the MaxSquares / Charbonnier loss for eg., as a fair comparison – including the diversity regularizer would be an even stronger baseline.
> >
> > 3) It is a little surprising that the batch version of the $L_{DIV}$ loss does so poorly even in comparison with the $L_{SLR}$ loss. A more detailed discussion would be useful.
> >
> > 4) While I understand that the choice may not be the optimal one, I'm still not entirely clear about the motivation behind the design of the ITM module (why is the convex combination necessary?) and think some additional context would be helpful. I would also recommend including a more detailed discussion of HLR v/s SLR in the paper along with recommendations/justifications of when it may be suitable (if ever) to use HLR over SLR.
> >
> > On the whole, I am slightly more positive about this paper now and have updated my rating to reflect this. I am still not convinced that the paper offers significant technical insights but rather combines methods from prior work effectively for the test-time setting, but am willing to reevaluate based on the author response.

---

> > > ### Author Response · Authors · 2021-08-24
> > > **We thank the reviewer for considering our response**
> > >
> > > 1) Thanks for pointing us to this related work. We will cite and discuss the similarities of the diversity regularizer with the SENTRY paper in our final version.
> > >
> > > 2) For a fair comparison to our work, we incorporated Charbonnier penalty and MaxSquares loss as drop-in replacements of the loss in our overall adaptation pipeline and hence adapted BN statistics as well as BN's affine parameters (as for the other losses). We did provide results of Charbonnier/MaxSquares with the diversity regularizer L_div in the above response.
> > >
> > > 3) Why does $L_{div}$ perform so poorly: When using only $L_{div}$, there is nothing that encourages predictions of the adapted network to stay 'close' to the predictions of the original network ($L_{div}$ is based on a network-independent prior $p_{D'}(y)$). In contrast, SLR corresponds to a type of self-supervision where the original network provides the initial self-supervision - and by this, the adapted network is encouraged to keep useful properties of the original network.
> > >
> > > 4a) why the convex combination in ITM: The convex combination is mainly helpful at initialization: setting $\tau=1$, $\gamma=1$ and $\beta=0$ as proposed in Section 3.1 leaves the input unchanged, that is: the ITM implements the identity. Without initializing $\tau=1$, a random initialization of $r_\psi$ would typically distort the input severely and the test-time adaptation would have to compensate for both the domain shift in input as well as the distortion caused by the initial $r_\psi$. By initializing $\tau=1$, ITM can more easily learn useful input transformation in $r_\psi$ from this overall well-behaved identity initialization.
> > >
> > > 4b) HLR vs. SLR: We will add a more detailed discussion of HLR versus SLR in the paper. We also refer to the additional results posted in our responses to Reviewer XAwq: (i) on a controlled 1d data experiment and(ii) an experiment on model predictions on a batch directly after it adapts on the batch. In both cases, SLR outperforms HLR consistently.
> > >
> > > Note that to the best of our knowledge, we are the first to motivate and identify the advantages of the HLR loss (that was proposed for supervised learning)  for self-supervised learning and test-time adaptation due to its non-saturating gradient property. Moreover, our work is the first to  propose the soft-label  SLR loss.
> > > Though there exist prior works that learns to transform inputs by an additional network module in different contexts, we would like to note that  our work differentiates itself from those prior work by learning this input transformation module in a fully self-supervised manner at test-time, and by this aids the adaptation.

---

### Official Review · Reviewer_XAwq · 2021-07-17

**Rating:** 6
**Confidence:** 4

**Summary:**

This paper presents a method for test time adaptation based on several techniques. These include a self-supervised adaptation objective based on log likelihood ratios, an additional regularizing objective to encourage diverse predictions, and an input transformation module that is also trained with the aforementioned objectives. Together, these techniques lead to better performance on ImageNet-C and ImageNet-R compared to Tent, a recent and similar test time adaptation method based on entropy minimization.

**Limitations And Societal Impact:**

Yes, the authors have done so adequately, though additional discussion would also be nice, even if just in the appendix. Note that this is a minor point and does not contribute to my overall score.

**Main Review:**

Originality
---
The paper itself notes that most of the techniques constituting the proposed method (diverse prediction regularization, input transformation module) are not novel. The self-supervised log likelihood ratio objective appears novel, as far as I am aware. I do not view the lack of originality as a substantial mark against this paper, as it is empirically focused and simply focuses on devising a better method for test time adaptation. And certainly, for this problem setting, the combination of the aforementioned techniques is novel and does lead to stronger empirical results than what has been previously reported.


Quality
---
The paper is of varying quality, and some improvements should be made as to this regard.

First, the paper could do a better job at motivating the proposed log likelihood ratio objective. Currently, there are two threads of motivation: intuitive, in the "non-saturating gradient" property, and empirical, in the improved results. But in their current form, these seem insufficient. For the intuitive motivation, it is not immediately clear why non-saturating gradients are desirable. If a model is already very confident about its prediction on a certain point, why and how should it derive any self-supervised signal to adapt? The idea behind entropy minimization methods like Tent is that confidence correlates with accuracy, so increasing confidence is desirable. This idea, of course, has flaws, but certainly it is intuitively understandable. It is not clear what the idea is behind the log likelihood ratio objective, or the general idea that adaptation can and should be performed when the model is confident. Furthermore, the gradient plot in Fig 1 does not seem to align well with the intuitive explanation for two reasons: the gradient of the entropy (green) is still non-trivially large for fairly high confidence, e.g., 0.9, and the caption says that entropy has "maximum gradient amplitude for low-confidence" prediction when this seems clearly false, e.g., the gradient around 0.5 is close to 0?

As for the empirical motivation that the objective simply performs better, this is nice but cannot fully substitute for better exposition of the idea itself. It is also unclear how important the input transformation module is, because I believe the paper does not include the baseline of Tent with this module (if I understand correctly, Tent+ only adds the diversity regularization). If such a baseline were to work very well, then it would turn out that the input transformation module is actually the most important piece, rather than changing the adaptation objective. Without results from this baseline, this possibility cannot be ruled out.

A more minor point is that it would be nice to include results in the setting where the model predicts on a batch directly after it adapts on the batch, rather than the current "offline" setting where several epochs are permitted on the entire test set before any predictions are made. In some practical applications, this would simply be infeasible, and predictions would have to be made right away. The paper could follow the general "online" setting that Tent uses.


Clarity
---
The paper is generally well written and structured. The main concern regarding clarity is that of the motivation behind the adaptation objective. Another more minor concern is whether the claim that the input transformation module can "undo distribution shifts" is justified, considering the mixed results in Table 2. Lastly, the exposition could be improved in 3.2.1 regarding the diversity regularization in running average mode. It seems like p_t ( y ) in L172 needs a stop gradient operator on p_t-1 ( y ), such that the gradients of the loss L_div in L173 are computed only with respect to p_t^emp. Otherwise, the method would involve storing a long history of model activations, and this seems unwieldy (and unlikely). If I am correct, it would be good to clarify this in the paper.


Significance
---
The paper focuses on an important topic (distribution shift) and an interesting emerging methodology (test time adaptation) with a moderate set of good results to support the proposed approach. Significance could be further improved through additional testbeds. For example, SVHN to MNIST/MNIST-M/USPS would offer another direct point of comparison to Tent. Alternatively, prior work [1] has shown that batch normalization based adaptation is ineffective on other test sets such as ImageNet-A and ImageNet-v2, so positive results on such test sets would be of high interest. The paper would be greatly strengthened through these additional experiments along with addressing the quality concerns noted above.

[1] Schneider et al, "Improving robustness against common corruptions by covariate shift adaptation". NeurIPS 2020.

---

Edit after reading the author response
---
As noted in my response below, my empirical concerns are resolved and I am thus upgrading my recommendation. I believe that, with the additional experiments, this work is above the bar for publication.

**Time Spent Reviewing:**

3

---

> ### Author Response · Authors · 2021-08-10
> **Thank you for the valuable feedback and raising interesting points. It further improves the understanding of our work.**
>
> **1) Motivation behind the log likelihood ratio adaptation objective:**
> The motivation for confidence maximization-type adaptation is that the decision boundary should be "far" from actual datapoints and the prediction confidence acts as a proxy for this distance. On the one hand, as the reviewer points out, the less confident the prediction on a datapoint, the closer is the datapoint to the decision boundary and the stronger the datapoint should be pushed  away from the boundary (large gradient). However, on the other hand, the lower the confidence, the less certain it becomes on which side of the decision boundary a datapoint should lie (what we call the "low confidence in self-supervision"), and thus the direction of the gradient becomes ambiguous. As evidenced by our empirical results, increasing the influence of high confidence predictions on the gradient as with our proposed losses improves performance. We argue that this is the case because it leads to better self-supervision (better gradient direction when averaged over the batch) than for the entropy loss, even though those highly confident predictions are already far from the decision boundary.
>
> **2) Clarification of Fig1 caption**
> A ResNet50 has a median prediction confidence of 0.929 on the clean ImageNet validation data. Because of this, we consider a confidence of 0.82, where the entropy's gradient gets maximal as "low". Moreover, a considerable higher than median confidence corresponds to 0.95+, where the entropy's gradient gets fairly small. We will clarify these notions of "low" and "high" confidence in our final version.
>
> **3) Experiment on input transformation module with TENT and TENT+:**
> The below table compares TENT and TENT+ with and without Input Transformation (IT) module on ResNet50 for all corruptions at severity level 5 adapted for 5 epochs. Reported are the mean accuracy(\%) across three random seeds (2020/2021/2022). Note that the error bars here are in a smaller scale or negligible as similar to the error bars reported in our appendix. While IT also improves performance when combined with TENT+, it is still clearly outperformed by SLR+IT.
>
> ||Gauss|Shot|Impulse|Defocus|Glass|Motion|Zoom|Snow|Frost|Fog|Bright|Contrast|Elastic|Pixel|JPEG|
> |:---------:|:---------:|:---------:|:---------:|:---------:|:---------:|:---------:|:---------:|:---------:|:---------:|:---------:|:---------:|:---------:|:---------:|:---------:|:---------:|
> |TENT|16.04|23.22|25.85|19.05|17.40|49.02|52.78|52.72|34.31|61.19|68.54|1.26|59.26|62.15|56.17|
> |TENT+IT|13.01|22.02|33.61|17.68|15.78|49.05|52.77|52.62|34.61|61.07|68.49|0.37|59.15|62.32|56.02
> |TENT+|33.97|37.95|36.93|32.69|33.36|51.42|54.33|54.55|45.80|62.09|69.03|24.08|60.36|63.10|57.21|
> |TENT++IT|35.27|39.47|42.80|32.21|33.33|51.44|54.16|54.48|46.32|62.07|69.04|21.31|60.24|63.32|57.30|
> |SLR|41.59|43.49|43.90|41.70|**41.10**|54.86|56.39|57.47|50.90|63.51|68.70|51.06|**61.36**|63.39|**58.35**|
> |SLR+IT|**43.09**|**44.39**|**64.05**|**41.98**|40.99|**55.73**|**56.75**|**58.56**|**51.68**|**63.64**|**68.85**|**55.01**|61.32|**63.59**|58.24|
>
> **Mixed results from input transformation (IT) module:**
> IT considerably increases accuracy on some distortions (for instance +20.0 percent points on Impulse,  +4.0 percent points on Contrast) and never reduces accuracy by more than 0.11 percent point. It improves performance on 11 out of the 15 distortion types. Thus the possible performance gains clearly outweigh the possible performance drops, and we would not call it a "mixed" result.
>
> **4) Experiment on model predictions on a batch directly after it adapts on the batch:**
> Results of different adaptation objectives with model predictions on a batch directly after adapting on the batch. Results are obtained with ResNet50 for all corruptions at severity 5 and reported mean accuracy(\%) across three random seeds. These numbers are obtained during the 1st epoch of adaptation but not after the 1st epoch of adaptation. We see that SLR still outperforms all the other methods in this setting.
>
> ||Gauss|Shot|Impulse|Defocus|Glass|Motion|Zoom|Snow|Frost|Fog|Bright|Contrast|Elastic|Pixel|JPEG|
> |:---------:|:---------:|:---------:|:---------:|:---------:|:---------:|:---------:|:---------:|:---------:|:---------:|:---------:|:---------:|:---------:|:---------:|:---------:|:---------:|
> |TENT|28.69|30.98|30.41|29.09|28.01|42.46|50.43|48.15|42.03|58.48|68.26|26.94|55.66|59.52|53.52|
> |TENT+|29.21|31.52|30.57|29.19|28.59|42.55|50.47|48.17|42.51|58.51|68.29|31.24|55.76|59.54|53.60|
> |HLR|33.11|35.99|35.15|33.34|33.35|46.61|52.97|51.55|45.67|60.10|68.04|42.97|57.97|60.40|55.07|
> |SLR|**35.04**|**37.86**|**37.08**|**35.22**|**35.12**|**48.50**|**53.74**|**52.89**|**46.72**|**60.75**|**68.32**|**44.91**|**58.86**|**61.21**|**55.90**|
>
> **5)  Clarification on the stop gradient operator on p_t-1 ( y ) in section 3.2.1:**
> We agree with the reviewer and we will add a "stop-gradient" operator around p_t-1(y). Thanks for pointing this out!
>
> **6) Experiments on SVHN to MNIST/MNIST-M/USPS:**
> ResNet26 is trained on SVHN dataset for 50 epochs with batch size 128, SGD optimizer with momentum 0.9 and initial learning rate 0.01, which drops to 0.001 and 0.0001 at 25th and 40th epoch respectively. ResNet26 obtains 96.49\% test accuracy on SVHN. Test time adaptation of SVHN trained ResNet26 to MNIST/MNIST-M/USPS is conducted with Adam optimizer with constant learning rate 0.001 for 20 epochs on TENT, TENT+ and SLR with three random seeds (2020/2021/2022). Note that the results of ResNet26 with TENT are not the same as the results reported in TENT [1], this might be attributed to the differences in training configurations of ResNet26 on SVHN, like optimizer, learning rate, learning rate scheduler, number of training epochs. These settings are not provided in the TENT [1] work, hence the resulted trained ResNet26 would provide different results. The table below compares our proposed loss SLR with TENT variants on ResNet26 and shows that our approach outperforms them across all the datasets.
>
> ||&nbsp;&nbsp;MNIST|||&nbsp;&nbsp;MNIST-M|||&nbsp;&nbsp;USPS|||
> |:--------|:------|:------|:------|:------|:------|:------|:------|:------|:------|
> |Epochs|1&nbsp;&nbsp;10&nbsp;&nbsp;20|||1&nbsp;&nbsp;10&nbsp;&nbsp;20|||1&nbsp;&nbsp;10&nbsp;&nbsp;20|||
> |TENT|85.6&nbsp;&nbsp;93.5&nbsp;&nbsp;93.5|||51.8&nbsp;&nbsp;56.8&nbsp;&nbsp;56.9|||80.4&nbsp;&nbsp;84.0&nbsp;&nbsp;84.0|||
> |TENT+|85.7&nbsp;&nbsp;96.3&nbsp;&nbsp;96.9|||53.6&nbsp;&nbsp;65.6&nbsp;&nbsp;67.4|||79.9&nbsp;&nbsp;84.7&nbsp;&nbsp;85.6|||
> |SLR(ours)|**86.9**&nbsp;&nbsp;**97.8**&nbsp;&nbsp;**98.3**|||**56.8**&nbsp;&nbsp;**74.9**&nbsp;&nbsp;**77.4**|||**80.3**&nbsp;&nbsp;**91.1**&nbsp;&nbsp;**94.2**|||
>
> Reported values are mean accuracy(\%) over three random seeds (2020/2021/2022). ResNet26 accuracy before adaptation on MNIST, MNIST-M and USPS are 42.48\%, 47.43\% and 11.83\% respectively.
>
> **7) Tests on ImageNet-A and ImageNet-v2:**
> We consider our method and also related work on test-time adaptation to be best suited when there is a shift in appearance in the inputs (distribution shifts involving changes in style and local image statistics). In other words: when the target distribution has support where the source distribution had none. ImageNet-A and -V2 do not fall into this category since they have very similar support to clean ImageNet but vary in the data subsampling, that is: they correspond to  different distribution with the same/very similar support (distribution shifts involving high-level and abstract changes). We can add this as an additional limitation of our work, albeit we do not consider it too limiting since appearance shift is a very frequent  issue in the real applications.

---

> > ### Comment · Reviewer_XAwq · 2021-08-15
> > **My empirical concerns are resolved, and thus my recommendation improves**
> >
> > Thanks for your response! I consider the empirical concerns that I raised in my initial review resolved, thus I am upgrading my recommendation by one point. I appreciate the new experiments on (and the clarifications for) the IT module, the new experiments in the online setting, and the new experiments on SVHN.
> >
> > 1) It would be nice to provide this argument more rigorously, which could be done in a number of ways. E.g., providing an illustrative example in which entropy minimization does the wrong thing (pushes ambiguous data points in the wrong direction) but the proposed objective is more successful. Or, some theoretical analysis for how a model behaves when the objective is fully optimized under appropriate assumptions. Basically, I am concerned that just making this argument more explicit in the paper will not be very convincing without some form of supporting evidence. Certainly, I would not be convinced.
> >
> > 2) Thanks for the clarification, though, it would be good to know whether or not a confidence of 0.82 is "low" for the corrupted image datasets or other instances of test distribution shift.
> >
> > 7) It would be nice to add this as an additional limitation of the work, though I also think it is fair to also mention in the paper that this is an inherited limitation of prior work. For what it's worth, even negative results on these datasets would still be interesting.

---

> > > ### Author Response · Authors · 2021-08-17
> > > **We thank the reviewer for taking our response into account**
> > >
> > > Thank you for providing valuable suggestions to further improve our work.
> > >
> > > * **Example supporting the motivation behind the log likelihood ratio adaptation objective:**
> > > We agree that providing an additional illustrative example would be helpful for supporting our proposed losses. We have devised the following simple 1D example, which supports our point further:
> > > Assuming  some (unlabeled) data that comes from the following bimodal distribution: $0.5\cdot \mathcal{N}(-1, 3) + 0.5\cdot \mathcal{N}(+1, 3)$, that is: half of the samples come from a normal distribution with mean -1 and the other half from a normal distribution with mean +1 (and both having standard deviation 3). We can interpret these two components of the mixture distributions as corresponding to data of two different classes, but class labels are of course unavailable during unsupervised test-time adaptation.
> > >
> > > We assume a simple logistic model of the form $p_\theta(y=1\vert x) = \frac{1}{1 + e^{-(x + \theta)}}$, where $x$ is the value of the data sample and $\theta$ is a scalar offset that determines the decision boundary. By construction, we know that the minimum density of the mixture distribution on $[-1, 1]$ is at 0. Since confidence maximization aims as moving the decision boundary to regions in input space with minimum data density (in this case to 0),  we can compare different self-supervised confidence maximization losses in the finite data regime as follows: for every finite data sample with $N$ data points $\\{x_i\\}$ for $i=1, \dots, N$ and  loss function $L$ , we  solve $\theta^*(L) = \arg\min_{\theta \in  [-1, 1]} L(\theta, \\{x_i\\})$, where the loss (such as entropy or SLR) is averaged over all data points. The absolute value $\vert  \theta^*(L) \vert$ gives us then an estimate of the error of the decision boundary parameter $\vert  \theta^*(L) \vert$ for the given data set and loss function.
> > >
> > >  Below, we provide this error for different loss functions and different number of data samples, averaged over 100 repetitions (shown are mean and standard error of mean):
> > >
> > > |#data|100|200|500|1000|2000|10000|20000|
> > > |:----:|:----:|:----:|:----:|:----:|:----:|:----:|:----:|
> > > |Entropy|0.487+-0.031|0.364+-0.029|0.230+-0.018|0.152+-0.013|0.117+-0.009|0.052+-0.004|0.033+-0.003|
> > > |MaxSquare|0.530+-0.032|0.421+-0.031|0.271+-0.022|0.179+-0.015|0.142+-0.011|0.061+-0.004|0.038+-0.003|
> > > |HLR|**0.357+-0.023**|**0.234+-0.018**|**0.145+-0.012**|**0.094+-0.008**|**0.071+-0.006**|**0.032+-0.002**|**0.022+-0.002**|
> > > |SLR|**0.332+-0.022**|**0.214+-0.017**|**0.140+-0.011**|**0.088+-0.008**|**0.067+-0.006**|**0.032+-0.002**|**0.021+-0.002**|
> > >
> > > It can be seen that SLR and HLR clearly outperform Entropy and the Max Squares loss for all data regimes. The difference between SLR and HLR is generally very small. While SLR seems to be consistently slightly better than HLR, this difference is not statistically significant. We attribute the superiority of SLR/HLR compared to entropy to the fact that all data points have non-saturating loss, regardless of their distance to the decision boundary. Thus, all data contributes to localizing the decision boundary, while for saturating losses such as the entropy, effectively only "nearby" points determine the decision boundary.
> > > We believe this example supports the point that our proposed non-saturating losses are beneficial over prior works for self-supervised confidence maximization.
> > >
> > > * **Tests on ImageNet-A:**
> > > We adapted ResNet50 on ImageNet-A dataset for 5 epochs with different losses as presented in the table below (reported is the model accuracy). We see that SLR outperforms the other losses. However, the improvements from the test-time adaptation on this dataset are minimal when compared to the results on corrupted datasets. Thus, test-time adaptation to be best suited when there is a shift in appearance in the inputs rather involving high-level and abstract changes as in ImageNet-A. As suggested by the reviewer, we will add this as an additional limitation of our work.
> > >
> > > ||No Adaptation|TENT|TENT+|HLR|SLR|
> > > |:----:|:----:|:----:|:----:|:----:|:----:|
> > > |ImageNet-A|0.0\%|0.04\%|0.08\%|0.51\%|**0.55\%**|

---

### Author Response · Authors · 2021-08-10
**We thank all the reviewers for providing valuable feedback**

We greatly appreciate that you find our work very clearly written [R3], well structured [R1], easy to follow [R2], with stronger empirical results than prior works [R1], and the improvements are convincing [R2]. As suggested, we conducted additional experiments including comparisons and clarification over the loss functions from previous works, and adaptation from SVHN to MNIST, MNIST-M, and USPS show the effectiveness of our proposed approach. We show that our approach is also effective when making the model prediction on a batch directly after it adapts on the batch. Furthermore, we clarified the motivation of our adaptation objective in more detail and extended the ablation studies. We will update our paper based on your suggestions for the final version accordingly. Below we respond to your main criticism and suggestions in detail.

---

### Decision · Program_Chairs · 2021-09-27

**Decision:**

Reject

**Comment:**

This paper presents a method for improving DNNs performance on distribution shifts via test time adaptation based on several techniques (a non-saturating loss function, diversity maximization and input transformation module). The paper requires significant modifications to be ready for publication in terms of clarity, additional comparison experiments, discussing the effectiveness of the method in more details, etc (see reviews and discussions for details). The discussions have not been enough to convince me and reviewers to accept the paper as is. I encourage authors to update the paper in light of discussions and feedback and submit to a future venue.